# The Association of Gut Microbiota and Complications in Gastrointestinal-Cancer Therapies

**DOI:** 10.3390/biomedicines9101305

**Published:** 2021-09-24

**Authors:** Kevin M. Tourelle, Sebastien Boutin, Markus A. Weigand, Felix C. F. Schmitt

**Affiliations:** 1Department of Anaesthesiology, Heidelberg University Hospital, 420, Im Neuenheimer Feld, D-69120 Heidelberg, Germany; kevin.tourelle@med.uni-heidelberg.de (K.M.T.); markus.weigand@med.uni-heidelberg.de (M.A.W.); 2Department of Infectious Disease, Medical Microbiology and Hygiene, Heidelberg University Hospital, 324, Im Neuenheimer Feld, D-69120 Heidelberg, Germany; sebastien.boutin@med.uni-heidelberg.de

**Keywords:** gastrointestinal cancer, colorectal cancer, gastric cancer, oesophageal cancer, microbiome, 16S rRNA, next-generation sequencing, complications, cancer therapy

## Abstract

The therapy of gastrointestinal carcinomas includes surgery, chemo- or immunotherapy, and radiation with diverse complications such as surgical-site infection and enteritis. In recent years, the microbiome’s influence on different diseases and complications has been studied in more detail using methods such as next-generation sequencing. Due to the relatively simple collectivisation, the gut microbiome is the best-studied so far. While certain bacteria are sometimes associated with one particular complication, it is often just the loss of alpha diversity linked together. Among others, a strong influence of *Fusobacterium nucleatum* on the effectiveness of chemotherapies is demonstrated. External factors such as diet or specific medications can also predispose to dysbiosis and lead to complications. In addition, there are attempts to treat developed dysbiosis, such as faecal microbiota transplant or probiotics. In the future, the underlying microbiome should be investigated in more detail for a better understanding of the precipitating factors of a complication with specific therapeutic options.

## 1. Introduction

In recent years, there has been increasing interest in the study of the microbiome. The microbiome of a 70 kg human is about 3.8 × 10^13^ bacterial cells, making up a total weight of about 200 g. In comparison, humans are composed of about 3.0 × 10^13^ human cells [1]. The microbiome comprises bacteria, viruses, fungi, bacteriophages, and protozoa. Those can interact in symbiosis or dysbiosis with their host. Different microbiomes can be investigated with different approaches and for various diseases. The colon, lungs, or the naso-oral area have different microbiome compositions. Many diseases show abnormalities in terms of an altered microbiome. The ability to study the microbiome in more detail using next-generation sequencing (NGS) provides new insight. NGS can analyse all DNA fragments, and thus identify and differentiate bacteria, fungi, and viruses from human DNA [2,3,4]. 

To date, no healthy baseline for a microbiome has been found. However, due to the many studies, we know that diversity and functional redundancy are key players to establish a balance in the gut microbiome [5]. Recent works also highlighted the impact of environmental factors such as diet, drugs, and anthropometric states rather than that of genetics in shaping the gut microbiome [6]. The Westernised diet and industrialization shift the gut microbiome to a pathobiont-orientated state by modifying the balance between Firmicutes and Bacteroidetes [5,7]. Those two phyla are mostly relevant due to their metabolic function and their complementarity in the production of short-chain fatty acids (SCFAs), which impact the human metabolism and immune system [8,9,10]. The imbalance of the gut microbiome, also called dysbiosis, induces a decrease in the control of the pathogens by commensals and by the immune system, which leads to a pathobiome [11]. Since every patient shows an individual microbiome, and the immune system and functional redundancy relate to strain-level resolution in function and immune recognition [5], a patient-specific baseline is required for a reliable assessment. To detect changes through interventions, the microbiome must be compared before and after an intervention. To avoid subsequent shifts after the intestinal microbiome is collected in the stool, the microbiome must be collected in specific preservation tubes and cooled as quickly as possible or preferably frozen at −80 °C until evaluation. The shorter the period between collection and preservation is, the more accurate results are [12,13]. 

The most common gastrointestinal cancers are oesophageal, gastric, and colorectal carcinomas. For all carcinomas globally, the colon carcinoma accounts for 6.1%, gastric carcinoma for 5.7%, rectal carcinoma for 3.9%, and oesophageal carcinoma for 3.2%. Of all cancers, 5.8% of patients died from colon cancer, 8.2% from gastric cancer, 3.2% from rectal cancer, and 5.3%% from oesophageal cancer [14]. Studies showed that certain bacteria and compositions of the microbiome may be responsible for the development of colorectal cancer. For example, *Escherichia coli* and *Bacteroides fragilis* were associated with earlier tumour onset and increased morbidity in mouse models [15]. In addition, *Fusobacterium nucleatum* is associated with tumorigenesis in colorectal cancer [16]. Furthermore, the microbiome differs in left- and right-side colorectal cancer [17,18]. The challenge of distinguishing whether certain bacteria have a selective advantage due to cancer, and are therefore more detectable in cancer patients, or whether certain bacteria promote cancer development remains unclear. 

In this review, we present the current status of the microbiome concerning therapeutic options for gastrointestinal carcinoma. More specifically, we show which complications are associated with the gut microbiome and, where possible, which bacterium, species, or phylum shows associated complications.

## 2. Materials and Methods

We completed a scientific search via PubMed in July 2021 to investigate microbiome-associated complications related to gastrointestinal-cancer treatment options. Human and animal studies, and studies with cell lines were included. Meta-analyses, RCTs, controlled trials, and case reports were also considered in this review. To screen relevant papers, “gastrointestinal cancer”, “gastric cancer”, oesophageal cancer”, and “colorectal cancer” combined with “surgery”, “chemotherapy”, “immunotherapy”, “radiation”, and combined with “complication”, “microbiome”, “microbiota”, “16S rRNA”, “metagenomic” and “next-generation sequencing” were used. Since there are few data on the microbiome of patients with colorectal cancer and radiation, and to give an overview of possible complications, additional data from cancer patients (prostate and cervical) who received pelvic radiation were used here. The main search period was from 2010 to July 2021. In the first step, suitable papers were selected by screening their abstracts, and if the content was suitable, the full text was examined. Papers not dealing with microbiome-associated complications related to gastrointestinal-carcinoma therapy were excluded. As a result, 36 papers were included in analysis (Table 1). All clinical data were entered into an electronic database (Excel 2017, Microsoft Corp., Redmond, WA, USA).

## 3. Gastrointestinal-Cancer Therapies and Microbiome

In gastric cancer, different tumour entities have different compositions of the gastric microbiome [48]. In colorectal carcinoma, there are microbiome changes compared to healthy individuals. In particular, Proteobacteria and Actinobacteria occur more frequently in patients with colorectal carcinoma [20]. In the following sections, we give an overview of the complications and corresponding microbiome for specific therapeutic options. Furthermore, in Figure 1 the key findings are summarized for a better overview.

### 3.1. Perioperative Complications 

Several studies showed that the microbiome is directly altered after surgery [21,22,23]. In abdominal surgery, anastomotic insufficiencies and abdominal infections are feared complications that often lead to life-threatening sepsis. The most common complications after colectomy in colorectal-cancer patients are anastomotic leakages (8.4%) and wound infection (13.4%) [49]. Patients with a colorectal carcinoma that showed reduced alpha diversity and a higher abundance of Lachnospiraceae were more prone to anastomotic insufficiencies [24]. A higher abundance of Bacteroidaceae is associated with an increased risk of anastomosis insufficiency [25]. In patients with colorectal carcinoma, abundant *Bifidobacterium* genus in the colorectal tissue is associated with an increased risk of anastomotic insufficiency [26]. A recent study from 2021 showed that rectal or colonic surgery could impact the microbiome for two years, and even after that time, the baseline was not reached. Diversity in patients with complications was lower than that in patients without complications [27].

After the resection of colorectal carcinoma, *Pseudomonas, Enterococcus*, *Staphylococcus*, and *Enterobacteriaceae* were significantly increased, and short-chain fatty acids (SCFAs) were significantly decreased [28]. Meta-analysis revealed that patients who had gastrointestinal surgery and higher diversity with more beneficial bacteria postoperatively also had a better outcome [50]. One study examined the skin microbiome after colorectal surgery and demonstrated a decrease in alpha diversity [29]. 

A further complication after colorectal surgery can be an ileus. Patients who had suffered a postoperative ileus had a significantly altered gut microbiome compared to patients who returned to normal bowel function [30].

In a study from Asia with 45 patients with oesophageal squamous cell carcinoma, *Streptococcus* and *Prevotella* spp. in the oesophageal tissue were independent risk factors for prognosis [31]. Patients with gastric cancer also experience changes in the microbiome during the perioperative period. Compared to the preoperative sample, genera *Escherichia/Shigella*, *Akkermansia*, *Dialister*, and *Lactobacillus* were more abundant [32].

In patients with gastric carcinoma, a shift in the gut microbiome was demonstrated in the perioperative setting. Thus, patients had fewer Bacteroides, and more *Escherichia/Shigella, Clostridium*, and *Veillonella* than healthy individuals did after gastrectomy. Patients also had a nonsignificant decrease in short-chain fatty acids in their stool after gastrectomy [32]. Furthermore, a more similar microbiome (lower beta diversity) between the oral and gastric microbiomes was associated with a lower risk of anastomotic leak after oesophageal resection [33]. 

### 3.2. Complications during Chemo- and Immunotherapy

Patients who suffer from gastrointestinal cancer often receive adjuvant or neoadjuvant chemotherapy, which alters the gut microbiome [35]. A retrospective study showed that patient outcomes can be improved during immunotherapy with bevacizumab using antibiotics. Patients with metastatic colorectal carcinoma had lower mortality rates if treated with antibiotics for a more extended period [36]. Another study showed that *Fusobacterium nucleatum* was abundant in colorectal-carcinoma patients with recurrence after chemotherapy. This study showed that *F. nucleatum* controls the Toll-like receptor, microRNA, and autophagy network, thus influencing cancer chemoresistance [37]. Among others, the *Fusobacterium nucleatum* bacterium is associated with a reduced response to chemotherapy [23,38]. 

Chemotherapy CapeOx includes capecitabine plus oxaliplatin. A study by Kong et al. showed that surgery and CapeOx chemotherapy significantly altered the gut microbiome, and may lead to an abundance of pathogenic bacteria [22]. Similar results were obtained in a study investigating chemotherapy-induced diarrhoea (CID) in patients with Stage III colorectal cancer after CapeOx chemotherapy and surgery. Here, a dysbiosis of the intestinal microbiome was found in CID patients compared to patients without CID. Among others, *Klebsiella pneumoniae* was most frequently detected in CID patients [13].

Another problem is the resistance of tumour cells to 5-fluorouracil (5-FU). *Fusobacterium nucleatum* also seems to play a role here by upregulating the expression of BIRC3 in tumour cells, thereby causing them to become insensitive to 5-FU [39]. In addition, *Fusobacterium nucleatum* is associated with oesophageal squamous cell carcinoma and cancer-specific survival [40], whereas probiotics, including *Bifidobacterium bifidum*, improve the outcome of 5-FU chemotherapy in rats with chemically induced colorectal cancer by enhancing the antitumour effect [41]. Carboxymethyl pachyman (CMP) is a polysaccharide that has anti-inflammatory and immune regulatory effects. In colorectal-carcinoma mice treated with 5-FU, the additional administration of CMP restored the diversity of the gut microbiome [42]. In patients with advanced gastric cancer, patients who had received neoadjuvant chemotherapy before surgery also had increased postoperative infections. Lower diversity and reduced *Bifidobacterium*, *Faecalibacterium*, and *Ruminococcus* in patients with postoperative infections were detected [43]. The next step is to modify the effects of *Fusobacterium nucleatum* to improve patient outcomes. A first study addressed this goal and demonstrated that Paris polyphyla, a herbal medicine, could inhibit both colorectal-cancer and *Fusobacterium nucleatum* growth in human colorectal-cancer cell lines [51].

Another complication of chemotherapy is mucositis, which is a dose-limiting factor. Thus, mucositis seems to be a complex interplay among the intestinal microbiome, the host cells, and the intestinal microenvironment. Unfortunately, only bacteria that promote mucositis, such as *Enterobacteriaceae*, and no protective bacteria were identified so far [52]. The administration of probiotics and omega-3 fatty acids in an RCT study of 140 patients benefited quality of life and side effects such as diarrhoea, nausea, and vomiting [44].

### 3.3. Complications during Radiotherapy

Radiotherapy can form part of the therapy against cancer. Complications associated with the patient’s intestinal microbiome may occur. In a study of patients with both colorectal adenoma and carcinoma, no difference was shown between patients who received surgery alone and patients who received chemotherapy or chemotherapy with radiation. However, the group of patients with colorectal carcinoma who received chemotherapy and radiation consisted of only 5 patients [45]. Thus, significantly more *Clostridium* IV, *Pascolarctobacterium*, and *Roseburia* were detected in prostate-cancer patients with radiation enteropathy. In addition, lower diversity was associated with radiation enteropathy [46]. Similar results were obtained in a study from 2019, which showed that cervical-cancer patients suffering from radiation enteritis had enriched *Coprococcus* in their gut microbiome before therapy. Patients suffering from radiation enteritis also had dysbiosis and lowered alpha diversity. At the same time, a lower abundance of Bacteroides, and a higher abundance of Gammaproteobacteria and Proteobacteria were present in patients with radiation enteritis [47]. 

## 4. Discussion

The microbiome impacts all therapeutic options in gastrointestinal carcinoma. It can influence the outcome of patients undergoing chemotherapy or immunotherapy and severely limit the wellbeing of patients who have undergone radiotherapy and suffer from mucositis or enteritis. Patients with decreased gut microbiome diversity show higher associated complication rates [53]. Furthermore, side effects can lead to dose limiting during chemotherapy. To adequately appreciate these effects in clinical practice, assessing the type of therapy and the underlying microbiome is crucial. Thus, classical therapy options could be adapted to the individual microbiome, and improve outcomes and quality of life during therapy. 

The optimisation of the gut microbiome impacts complications. *Enterococcus faecalis* is associated with an increased rate of anastomosis insufficiency, and second- and third-generation cephalosporins failed to eradicate *E. faecalis* from anastomotic tissue [54]. In addition, there was a positive effect on anastomosis healing by a low-fat/high-fibre diet in a murine model, which was attributed, among other things, to an improved diversity of the intestinal microbiome [55]. The positive effect of SCFAs was also demonstrated in animal models. With the administration of SCFAs, the strength of anastomosis could be significantly increased compared to in the group that did not receive SCFAs [56,57]. In a mouse model, an altered gut microbiome could also lead to surgical-site infection. Surgical-site infection could be caused by the spread of bacteria by immune cells [58]. Whether therapy with probiotics can positively influence surgical-site infections is not yet conclusively clarified [59]. According to the ERAS protocol, patients who had received perioperative immunonutritrion had significantly fewer infectious complications, such as surgical-site infections [60]. In addition, next-generation sequencing can help to identify culture-negative surgical-site infections [61]. 

Firmicutes/Bacteroides ratio (F/B ratio) is also a well-studied marker for dysbiosis, and is altered during the treatment of gastrointestinal carcinoma. If the F/B ratio of intensive-care patients is <0.1 or >10, they have a poorer outcome [62]. Thus, patients who had received pelvic radiation, including for colorectal or anal carcinoma, experienced radiation-induced diarrhoea with an increased F/B ratio and dysbiosis compared with patients who did not experience diarrhoea [63]. Thus, a goal of future therapy is that the F/B ratio improves again after treatment. For example, the F/B ratio is increased in obesity [64,65]. At the same time, sport leads to a reduction in F/B ratio [66]. Whether exercise also positively impacts complication rates in the therapy of gastrointestinal carcinoma is unknown. However, no studies demonstrated any effect on the F/B ratio in dysbiosis [12,67]. The extent to which the F/B ratio is also an indicator of dysbiosis in the treatment of gastrointestinal carcinoma remains to be investigated. 

One way to reduce complications is to actively influence the microbiome. Probiotics are beneficial bacteria that have a positive effect on the gut microbiome. In comparison, prebiotics are dietary fibres that help beneficial bacteria to grow. Synbiotics consist of prebiotics and probiotics. A study with colorectal-carcinoma patients who had undergone surgery and received probiotics showed that they had significantly fewer days to their first postoperative defecation and flatus than the placebo group did. Diarrhoea was also less frequent in the probiotic treatment group [68]. In a cohort with 46 patients undergoing resection for periampullary neoplasm, a significant reduction in mortality, duration of antibiotic therapy, length of hospital stays, infections, and postoperative complications was shown with the administration of synbiotics [69]. A review investigated the effect of probiotics and synbiotics in colorectal patients undergoing surgery. There were significantly less severe diarrhoea, postoperative infections, chemotherapy side effects, and a shorter duration of antibiotic therapy in patients treated with probiotics or synbiotics [70]. In the colorectal-carcinoma mouse model, probiotics containing *Pediococcus pentosaceus* inhibit tumour growth and modulate gut microbiome dysbiosis towards eubiosis [71]. Cochrane analysis could not demonstrate a relevant effect of probiotics on radiotherapy- or chemotherapy-induced diarrhoea [72]. 

Numerous factors can influence the microbiome, e.g., diet, obesity, and physical activity. An increased BMI during laparoscopic colorectal-cancer surgery [73] and preoperative obesity is associated with poorer outcomes. On the other hand, early oral nutrition and thereby a more physiological gastrointestinal microbiome are associated with better outcomes [74]. After gastrectomy, 41.5% of patients suffer from sarcopenia, which highlights the importance of a sufficient nutritional status [75]. That diet and nutritional status affect the microbiome also shows that decreased secondary bile acids in the gut are a reasonable basis for the growth of *Clostridium difficile* [76]. A study with mice suffering from multiple sclerosis showed that one hour of strength exercise increased alpha diversity and abundance of the gut microbiota. Additionally, the abundance of short-chain fatty acid-producing bacteria and the F/B ratio were decreased [77]. Whether strength exercise has a similar effect on the microbiome in gastrointestinal carcinoma patients remains unclear.

Another complication of chemotherapy with irinotecan in patients with colorectal carcinoma is diarrhoea. The gut microbiome decomposes irinotecan–metabolite back to the active form, causing side effects. A new therapy, together with apple pectin, was able to inhibit metabolites responsible for diarrhoea in cell culture with human colorectal tumour cells and at the same time enhance the cytotoxic effect of irinotecan [78]. However, nonresponse to chemotherapy may also represent a complication of chemotherapy. New therapeutic approaches try to inhibit the growth of *F. nucleatum* by phage-guided nanotechnology and improve the first-line chemotherapy treatment [79]. Furthermore, the fact that the side effects of chemotherapy are often treated with other drugs should not be ignored. Thus, animal models showed that proton pump inhibitors, opioids, and antipsychotics are associated with dysbiosis [80]. In patients, opioids, and antibiotics such as piperacillin/tazobactam, are associated with alterations of the gut microbiome, and smaller amounts of protective bacteria such as *Blautia* and *Lactobacillus* are detectable [81]. Nevertheless, the preoperative administration of oral antibiotics could lead to significantly fewer postoperative complications in elective colorectal surgery than those in mechanical bowel preparation [82].

Another therapeutic option could be faecal microbiota transplant (FMT) to influence the microbiome in patients with gastrointestinal cancer under specific therapies. Currently, data on the effect of FMT in gastrointestinal carcinoma is insufficient. However, animal models showed that, when the FMT donor was a patient with colorectal carcinoma and the FMT recipient mouse was germ-free, intestinal dysplasia and proliferation could be induced [83]. In another colorectal-carcinoma mouse model, oral–faecal microbiota transplantation was performed during five days of FOLFOX chemotherapy and two days beyond,§ and compared to mice that did not receive FMT. Mice treated with FMT had less severe diarrhoea and reduced mucositis, and at the same time, it did not induce bacteraemia. In addition, dysbiosis was improved in FMT-treated mice after FOLFOX chemotherapy [84]. The microbiome and health status of the patient can thus be significantly changed by FMT. Whether physiological conditions can also be achieved by FMT is unknown for gastrointestinal carcinoma. However, FMT should not be underestimated, as shown by case reports in which patients had suffered severe complications or died due to FMT [85,86,87,88]. Nevertheless, FMT is a promising therapeutic option, but further studies are needed to evaluate it and identify its risks leading to serious side effects. 

A feared complication in patients undergoing surgery on the gastrointestinal tract is sepsis, even though these are not specific complications for operations on the gastrointestinal tract. The gut microbiome also interacts with the immune system. Therefore, an altered microbiome cannot be neglected concerning the immune response. This directly influences the mucosal adaptive immunity and immune cells [89]. The microbiome influences the course of sepsis [90]. Patients who have undergone gastrointestinal surgery and suffer from dysbiosis resulting from the procedure may have a worse outcome if they develop sepsis. 

Unfortunately, there are currently insufficient data for patients with gastrointestinal carcinoma under radiation and possible complications associated with the microbiome. Therefore, additional studies with prostate- and cervical-cancer patients were presented here. Furthermore, dysbiosis and associated inflammation were found with pelvic radiation in mice, which may be an additional factor for intestinal radiation-induced damage [91].

Furthermore, the microbiota does not only consist of bacteria; viruses and fungi are an additional part of it. The virome as part of the microbiota also impacts outcomes in patients with colorectal cancer. Thus, virome-associated risk groups could be identified [92]. The extent to which individual viruses, fungi, and bacteria affect both the patient and each other must be investigated in more detail.

Another challenge is that the microbiome differs depending on whether a healthy organ or a carcinoma is present, and the type of underlying carcinoma. For example, different microbiomes could be detected between oesophageal squamous cell carcinoma tissue and gastric cardia adenocarcinoma tissue [93]. This was confirmed in a microbiome comparison of a healthy oesophagus, Barrett’s oesophagus, an oesophageal squamous cell carcinoma, and oesophageal adenocarcinoma [94]. 

The microbiome in colorectal carcinoma is better studied compared to oesophageal and gastric carcinomas. So far, data for oesophageal and gastric carcinomas are limited regarding perioperative complications and complications during chemotherapy. Thus, only statements about the overall outcome or risk of recurrence with different microbiomes in oesophageal carcinoma have been made so far [31,95].

## 5. Conclusions

Therapeutic options of gastrointestinal carcinoma are surgery, chemo- or immunotherapy, and radiation. Each therapy option can cause different complications, which can be dose-limiting. Through next-generation sequencing, the underlying microbiome during a complication could be studied in more detail. This has allowed for the investigation of risk constellations, such as the presence of abundant *Fusobacterium nucleatum* during chemotherapy. Unfortunately, to date, only the gut microbiome in colorectal cancer and associated complications has mainly been investigated. Future therapeutic approaches should consider any individual differences in the underlying microbiome. Accordingly, the microbiome during peritherapeutic complications should be further investigated for all gastrointestinal carcinomas.

## Figures and Tables

**Figure 1 biomedicines-09-01305-f001:**
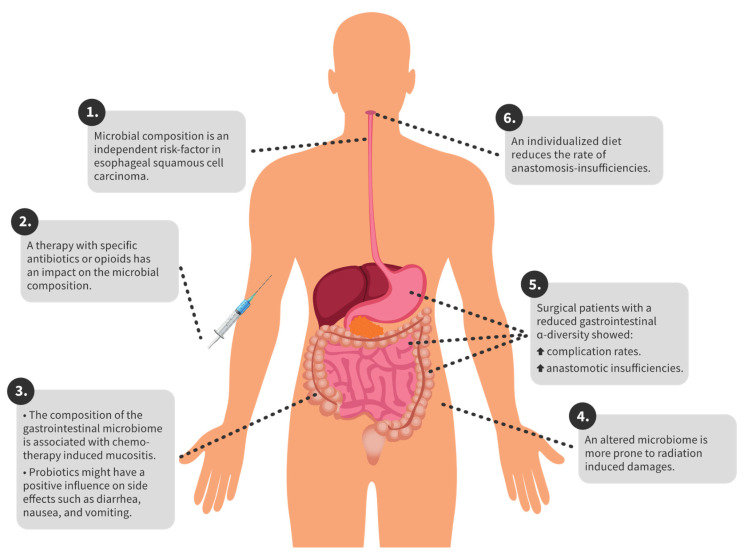
Overview of key findings.

**Table 1 biomedicines-09-01305-t001:** Study characteristics of included original papers. AL—anastomotic leakage, CID—chemotherapy-induced diarrhoea, CRC—colorectal carcinoma, GC—gastric cancer, RE—radiation enteropathy, RCT—randomized control trial, SSI—surgical site infection.

Author	Year	Study Characteristics	Population	Sample Type	Methods	Results
Liu, X et al. [19]	2019	Retrospective cohort trial	Human	Gastric and tumoral tissue	276 patients with gastric cancer without neoadjuvant chemotherapy who received surgery microbial diversity between normal (230) vs. Peritumoral (247) vs. tumoral (229) tissue	Decreased richness in peritumoral and tumoral tissue Increased abundance of *Streptococcus anginosus*, *Propionibacterium acnes*, *Prevotella melaninogenica* in tumor microbiota Decreased abundance of *Bacteroides uniformis*, *Helicobacter pylori* (HP) and *Prevotella capri* in tumor microbiota Positive patients had higher gastric acid secretion In normal and peritumoral microhabitants, a colononisation with high abundance of HP influences the overall composition of the microbiota in the stomach
Sinha, R. et al. [20]	2016	Case–control trial	Human	Stool samples	42 samples from CRC patients vs. 89 matched controls (elective surgery but no cancer or gastrointestinal conditions)	Higher abundance of *Fusobacterium* and *Porphyromonas* in CRC Lower abundance of Clostridia, Lachnospiraceae in CRC
Okazaki M. et al. [21]	2013	RCT	Human	Stool samples	48 patients over 70 years who received surgery S group: synbiotic therapy 7 days preoperative and 10 days postoperative (25 patients) C group: no synbiotic therapy (23 patients)	More short-chain fatty acids and lower pH in the S group S group: lower abundance of Enterobacteriaceae, *Pseudomonas* and *Staphylococcus* and stable abundance of *Bifidobacterium* and *Lactobacillus* Postoperative infections: S group (12%) vs. C group (36%) (*p* = 0.06)
Kong, C. et al. [22]	2019	Observational trial	Human	Stool samples	43 CRC patients who received surgery and adjuvant chemotherapy	Adjuvant chemotherapy alters the intestinal microbiota with higher abundance of pathogenic bacteria After surgery, higher abundance of *Veillonella*, *Morganella*, *Streptococcus*, *Proteus*, *Escherichia/Shigella*, *Blautia* and Enterobacteriaceae After surgery, lower abundance of *Bacillus*, *Bilophila*, *Enterococcus* and *Barnesiella*
Deng, X. et al. [23]	2018	Control trial	Human	Stool samples	69 patients Group C: healthy individuals (33) Group I: samples before treatment of CRC patients (17) Group IO: surgically treated CRC patients (5) Group D: chemotherapeutically treated CRC patients (14)	Reduced alpha-diversity in surgically treated patients Higher abundance of *Fusobacterium nucleatum*, *Veillonella dispar* and *Sutterella* in chemotherapeutically treated patients -> potential chemoresistance
van Praagh, J.B. et al. [24]	2016	Control trial	Human	Colorectal tissue	16 patients who received colorectal surgery 8 patients who received AL vs. 8 matched patients with no AL	Higher abundance of Lachnospiraceae in AL Higher BMI possibly associated with higher abundance of Lachnospiraceae Higher alpha diversity in patients without AL
van Praagh, J.B. et al. [25]	2019	Control trial	Human	Colorectal tissue	122 patients who received colorectal surgery	Development of AL was associated with a high abundance of Bacteroidaceae and Lachnospiraceae and a low alpha-diversity in patients with a non-C-seal anastomosis AL rates were higher in C-seal anastomosis patients (25% vs. 17%) Patients with C-seal anastomosis and AL had lower abundance of *Prevotella oralis*
Mima, K. et al. [26]	2020	Observational trial	Human	Colorectal tissue	256 samples from patients with colorectal cancer	Tissue with higher abundance of *Bifidobacterium genus* had an increased risk for AL (multivariable odds ratio 3.96) No association of AL and a high abundance of *Fusobacterium nucleatum*, *Enterococcus faecalis* and *Escherichia coli*
Schmitt, F.C.F. et al. [27]	2021	Control trial	Human	Stool samples	62 patients with colorectal cancer 30 patients with postoperative complications vs. 32 without postoperative complications	Reduced alpha diversity in patients with postoperative complications, and did not reach their preoperative microbiota composition even after 24 months after surgery Significant altered microbiome in patients with postoperative complications 6 month after surgery compared to their preoperative baseline -> associated with long-lasting higher abundance of *Fusobacterium ulcerans*
Ohigashi, S. et al. [28]	2013	Observational trial	Human	Stool samples	81 patients with colorectal cancer (comparison of the preoperative and postoperative microbiota)	Lower amounts of short-chain fatty acids after surgery Postoperative higher abundance of Enterobacteriaceae, *Pseudomonas*, *Staphylococcus* and *Enterococcus* Postoperative decreased abundance of obligate anaerobs like *Clostridium coccoides* group, *Bacteroides fragilis* group, Prevotella, *Atopobium* group, *Clostridium leptum* subgroup and *Bifidobacterium* and lower total bacterial counts
Holder-Murray, J. et al. [29]	2020	Observational trial	Human	Stool samples and skin tissue	60 patients who receive colorectal surgery	Decreased alpha-diversity of the skin at the surgical site after surgery One patient with a higher abundance of *Enterococcus* received a wound infection At the surgical site was a transient postoperative loss of *Propionibacterium* and *Corynebacterium*
Shogan, B.D. et al. [30]	2020	Observational trial	Human	Rectal sample	101 patients who received colon or rectal surgery	Significant altered microbiome between the preoperative and postoperative samples in patients with postoperative ileus compared to patients who returned to normal bowel function higher abundance of *Bacteroides* spp., *Ruminococcus* spp. and *Parabacteroides* spp. in patients with postoperative ileus patients with SSI or AL had no difference in their microbiome
Liu, Y et al. [31]	2018	Observational trial	Human	Oesophageal tissue	45 patients with oesophageal carcinoma and oesophageal resection	Patients with lymph-node metastasis had higher abundance of Firmicutes, Bacteroidetes, Spirochaetes and genera *Prevotella* and *Treponema*, and lower abundance of Proteobacteria Patients with T3–4 stage oesophageal carcinoma had higher abundance of *Streptococcus* compared to T1–2 Multivariant analysis showed unfavourable survival with combined high abundance of *Streptococcus* and *Prevotella*
Liang, W. et al. [32]	2019	Control trial	Human	Stool samples	20 patients with GC and surgery vs. 22 healthy controls	Compared to healthy individuals the patients with GC had lower abundance of *Bacteroides* and higher abundance of *Escherichia/Shigella*, *Veillonella* and *Clostridium XVIII* GC patients had higher abundance of *Dialister*, *Escherichia/Shigella*, *Prevotella* and *Akkermansia* after their surgery compared to the preoperative samples
Reddy, R.M. et al. [33]	2018	Observational trial	Human	Oral saliva, oesophageal and gastric tissue, neck swab or sputum samples	55 patients which received transhiatal esophagectomy (51 patients had a carcinoma)	No correlation between the gastrointestinal microbiome and the carcinoma Patients with anastomotic leakage after esophagectomy correlates with higher variance in the preoperative microbiome of the stomach and the oral cavity
Kumar, A. et al. [34]	2011	Case report	Human	Blood samples	48 year old patient with adenocarcinoma of the gastro-oesophageal junction who received total parenteral nutrition after surgery	Bacteremia with *Weissella confusa* confirmed with blood cultures and 16S rRNA sequencing
Shuwen, H. et al. [35]	2020	Control trial	Human	Stool samples	11 CRC patients treated with chemotherapy (FOLFIRI) vs 15 CRC patients treated with chemotherapy (XELOX) and after surgery vs. CRC patients without chemotherapy or surgery	CRC patients treated with XELOX chemotherapy had higher abundance of *Humicola*, *Veillonella*, Tremellomycetes and *Malassezia* CRC patients treated with FOLFIRI chemotherapy had lower abundance of Clostridiales, *Phascolarctobacterium*, *Humicola*, *Faecalibacterium* and *Rhodotorula* and higher abundance of Dipodascaceae, Tremellomycetes, *Candida*, Magnusiomyces, *Saccharomycetals*, *Lentinula* and *Malassezia* If patients treated with FOLFIRI chemotherapy received additionally cetuximab, higher abundance of *Candida*, Tremellomycetes, *Lentinula*, *Malassezia*, Dipodascaceae, and Saccharomycetales, and lower abundance of *Rhodoturola*, Magnusiomycetes and *Humicola* were found compared to patients treated with FOLFIRI chemotherapy alone
Lu, L. et al. [36]	2019	Retrospective cohort trial	Human	-	147 patients with metastatic colorectal cancer (mCRC) who were treated with bevacizumab (association between outcome and antibiotic use)	Patients with mCRC treated with bevacizumab and received a longer period of antibiotics could be associated with a better outcome
Yu, T. et al. [37]	2017	Control trial	Mouse and cell lines and bacterial Strains, and human	Numerous	48 CRC patients with recurrence (tissue) vs. 44 CRC patients without recurrence Cultivation of numerous bacterial strains Cultivation of numerous cell lines, some with oxaliplatin to establish resistance Numerous mouse models with injected Fusobacterium nucleatum and some models with different doses of oxaliplatin treatment	After a chemotherapy a colorectal carcinoma recurrence is associated with parcticular abundant bacteria CRC patients may benefit to treat *Fusobacterium nucleatum* colonisation in terms of management and prognosis
Yan, X et al. [38]	2017	Control trial	Human	Colorectal tissue	280 Stage III/IV CRC patients underwent surgery Comparison of tumor vs. adjacent normal tissue	Tumour stage, lymph node and distant metastasis correlated with high abundance of *Fusobacterium nucleatum* *Fusobacterium nucleatum* further was independent adverse prognostic factor for survivial CRC patients with lower abundance of *Fusobacterium nucleatum* benefit more than CRC patients with higher abundance of *Fusobacterium nucleatum* in terms of adjuvant chemotherapy
Fei, Z. et al. [13]	2019	Control trial	Human	Stool samples	4 stage III CRC patients with CID vs. 13 stage III CRC patients without CID all included patients received CapeOx chemotherapy (capecitabine and oxaliplatin)	Lower diversity and richness in patients with CID Higher abundance of *Klebsiella pneumoniae* in stage III CRC patients and CID
Zhang, S. et al. [39]	2019	Control trial (mice were randomized)	Cell lines (human and mouse)	Cell lines and CRC tissue	Study of the expressed genes induced by *Fusobacterium nucleatum* in CRC cell lines 20 mice in 4 groups for different experiments	*Fusobacterium nucleatum* induced the expression of the BIRC3 gene higher 5-FU chemoresistance of CRC cell lines infected with *Fusobacterium nucleatum* High abundance of *Fusobacterium nucleatum* in CRC patients after radical surgery who received 5-FU chemotherapy
Yamamura, K. et al. [40]	2016	Control trial	Human	Oesophageal cancer tissue	325 esophageal cancer patients underwent surgery 74 esophageal cancer patients with Fusobacterium nucleatum positiv tissue vs. 251 esophageal cancer patients with Fusobacterium nucleatum negativ tissue	Tumour stage was associated with *Fusobacterium nucleatum*
Genaro, S.C: et al. [41]	2019	RCT	Rats	Colon tissue	25 male rats 5 healthy rats without CRC vs. 5 CRC rats without treatmet vs. 5 CRC rats received 10 weeks 5-FU vs. 5 CRC rats received 10 weeks of 5-FU + probiotic vs. 5 CRC rats received 10 weeks of probiotics	Two rat groups that had received probiotics had lower aggressiveness of the CRC, meassured with the count of aberrant crypt foci and lower proportion of malignant neoplastic lesions
Wang, C. et al. [42]	2018	RCT	Mouse cell lines	Colon tissue and stool samples	50 mice 10 healthy mice without CRC vs. 10 CRC mice without treatment vs. 10 CRC mice treated with 5-FU vs. 10 CRC mice treated with 50mg*kg-1 carboxmethylated pachyman (CMP) + 5-FU vs. 10 CRC mice treated with 100mg*kg-1 CMP + 5-FU	Lower alpha diversity in the CRC mice compared to the healthy group CRC mice had higher abundance of Firmicutes and Preoteobacteria and lower abundance of *Bacteroides* compared to the healthy group Higher alpha diversity in CRC mice treated with 5-FU and CMP compared to CRC mice treated only with 5-FU Combination of CMP with 5-FU rever reversed the shortening of the intestine
Wei, Z. et al. [43]	2017	Control trial	Human	Jejunum tissue	90 GC patients underwent surgery 60 patients underwent srugery (SURG group) vs. 30 patients underwent neoadjuvant chemotherapy and surgery (NACT group)	In the NACT group, more infectious complications (disrupted tight junctions, downregulated claudin-1, ZO-1 and occludin) Decreased alpha diversity in NACT group Lower abundance of *Ruminococcus*, *Faecalibacterium* and *Bifidobacterium* in NACT group
Golkhalkhali, B. et al. [44]	2018	RCT	Human	Blood samples	140 CRC patients on XELOX chemotherapy (capecitabine and oxaliplatin) 70 CRC patients received microbial cell preperation (MCP) and omega-3 fatty acids vs. 70 CRC patients received placebo	Fewer side effects of chemotherapy and improved quality of life in the intevention group Rising levels of TNF-alpha and IL-6 in the placebo group Reduced IL-6 levels in the intervention group
Sze, M.A. et al. [45]	2017	Control trial	Human	Stool samples	67 patients 22 patients with colorectal adenoma vs. 19 patients with colorectal advanced adenoma vs. 26 patients with colorectal carcinoma	No differences in patients microbiota treated with only surgery compared to chemotherapy or chemotherapy and radiation Treatment of the microbiota of carcinoma patients underwent more alterations than those of the microbiota of adenoma patients.
Reis Ferreira, M. et al. [46]	2019	Control trial	Human	Stool samples and intestinal mucosa	134 prostate cancer patients after radiation 32 patients assesed for early radiation enteropathy vs. 87 patients assesed for late radiation enteropathy vs. 9 prostate cancer patients with radiation therapy received a coloscopy vs. 6 healthy controls who received a coloscopy	Patients with a higher alpha diversity before radiation therapy had no symptoms of enteropathy Higher abundance of *Roseburia*, *Phascolarctobacterium* and *Clostridium IV* in patients with radiation enteropathy Reduced Interleukins (IL7, IL12/IL23p40, IL15, IL16) in the intestinal mucosa of patients suffering from radiation enteropathy
Wang, Z. et al. [47]	2019	Control trial	Human	Stool samples	18 cervical-cancer patients (Stages II–IV) 10 patients with RE vs. 8 patients without RE	Patients suffer from RE had a lower alpha diversity but higher beta diversity Lower abundance of *Bacteroides* and a higher abundance of Gammaproteobacteria and Proteobacteria Patients with higher richness of *Coprococcus* before radiation therapy were more likely to develop RE Patients suffering from mild RE had higher abundance of *Alcanivorax* and *Virgibacillus*

## Data Availability

Not applicable.

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
