# Peer review of "The Association of Gut Microbiota and Complications in Gastrointestinal-Cancer Therapies"

_biomedicines, 2021, doi:10.3390/biomedicines9101305_

Round 1
Reviewer 1 Report
The relationship between the human gut microbiome and health is a fascinating topic of study. The authors propose an interesting and in-depth review on the association of gut microbiota and complications in gastrointestinal cancer therapies. However, this reviewer would like some modifications to be made before approval.
- The human gut microbiome is increasingly being studied and as the authors pointed out, no healthy baseline for a microbiome has been found. However, it will benefit this review if the authors could expand a bit this review and comment on the latest work done and finding toward defining a healthy microbiome. It will help the reader to be able to compare and have a better grasp on the alteration of the microbiome in gastrointestinal cancers.
- It feels like the abstract was quickly written and, in its current form, doesn’t fully match the content of the manuscript.
- The discussion section has some redundancy and needs to be streamlined and re-organized for clarity.
Author Response
Dear Reviewer,
Please find enclosed the revised version of the manuscript entitled “The association of gut microbiota and complications in gastrointestinal cancer therapies” and a detailed point-to-point response to the reviewers’ comments.
We hope you will find the revised version appropriate for publication in Biomedicines.
On behalf of all authors,
Yours sincerely,
Felix Schmitt, MD
Response to the Reviewers
Response to the suggestions of Reviewer 1:
- Reviewer’s suggestion:
The human gut microbiome is increasingly being studied and as the authors pointed out, no healthy baseline for a microbiome has been found. However, it will benefit this review if the authors could expand a bit this review and comment on the latest work done and finding toward defining a healthy microbiome. It will help the reader to be able to compare and have a better grasp on the alteration of the microbiome in gastrointestinal cancers.
Authors’ statement:
We agree with the reviewer that the healthy microbiome should also be further addressed for a better overview. Therefore, we have implemented another paragraph elaborating the current knowledge about a healthy microbiome and influencing factors on a healthy microbiome.
Related text section:
Introduction
To date, no healthy baseline for a microbiome has been found. However, due to the many studies, we do know that diversity and functional redundancy are key-players to establish a balance in the gut microbiome [5]. Recent works have also highlighted the impact of environmental factors such as diet, drugs, and anthropometric states rather than genetic in shaping the gut microbiome [6]. The westernized diet and industrialization have been shown to shift the gut microbiome to a pathobiont oriented state by modifying the balance between Firmicutes and Bacteroidetes [5,7]. Those two phyla are mostly relevant due to their metabolic function and their complementarity in the production of short-chain fatty acids (SCFAs) which are known to impact human metabolism and immune system [8-10]. The unbalance of the gut microbiome also called dysbiosis induces a decrease in the control of the pathogens by commensals and by the immune system which leads to a pathobiome [11]. Since every patient shows an individual microbiome and that the immune system and functional redundancy relate to strain-level resolution in the function and immune recognition [5], a patient-specific baseline is required for a reliable assessment. To detect changes through interventions, the microbiome must be compared before and after an intervention. To avoid subsequent shifts after the intestinal microbiome has been collected in the stool, the microbiome must be collected in specific preservation tubes, and cooled as quickly as possible, or, preferably, frozen at -80°C until evaluation. The shorter the period between collection and preservation, the more accurate the results [12,13].
- Reviewer’s suggestion:
It feels like the abstract was quickly written and, in its current form, doesn’t fully match the content of the manuscript.
Authors’ statement:
According to the Reviewers suggestion, we have completely rewritten the abstract to provide a more appropriate overview of the underlying review.
Related text sections:
Abstract:
The therapy of gastrointestinal carcinomas include surgery, chemo- or immunotherapy, and radiation with diverse complications like surgical site infection, enteritis, and others. In recent years, the microbiome's influence on different diseases and complications has been studied in more detail using methods such as next generation sequencing. Due to the relatively simple collectivization, the gut microbiome is the best-studied so far. While sometimes certain bacteria are as-sociated with one particular complication, it is often just the loss of alpha diversity linked together. Among others, a strong influence of Fusobacterium nucleatum on the effectiveness of chemo-therapies could be demonstrated. External factors such as diet or specific medications can also predispose to dysbiosis and consequently lead to complications. In addition, there are attempts to treat a developed dysbiosis, such as fecal microbiota transplant or probiotics. In the future the underlying microbiome should be investigated in more detail, for a better understanding of the precipitating factors of a complication with specific therapeutic options.
- Reviewer’s suggestion:
The discussion section has some redundancy and needs to be streamlined and re-organized for clarity.
Authors’ statement:
We appreciate the reviewer's feedback. We have restructured and better organized the discussion to avoid redundancies.
Related text sections:
Discussion:
The microbiome has an impact on all therapeutic options in gastrointestinal carcinoma, as shown above. It can influence the outcome of patients undergoing chemotherapy or immunotherapy and severely limit the wellbeing of patients who have undergone radiotherapy and suffer from mucositis or enteritis. Patients with decreased gut microbiome diversity show associated higher complication rates [53]. Furthermore, side effects can lead to dose-limiting during chemotherapy. To adequately appreciate these effects in clinical practice, assessing the type of therapy and the underlying microbiome is crucial. Thus, in the future, classical therapy options could be adapted to the individual microbiome and improve the outcome and quality of life during therapy.
As mentioned, the optimisation of the gut microbiome has an impact on complications. Enterococcus faecalis was shown to be associated with an increased rate of anastomosis insufficiency, and 2nd and 3rd generation cephalosporins failed to eradicate E. faecalis from anastomotic tissue [54]. In addition, a positive effect on anastomosis healing by a low fat/high fibre diet was shown in the murine model, which was attributed, among other things, to an improved diversity of the intestinal microbiome [55]. The positive effect of SCFAs was also demonstrated in animal models. With the administration of SCFAs, the strength of anastomosis could be significantly increased compared to the group that did not receive SCFAs [56,57]. In a mouse model, it was also demonstrated that an altered gut microbiome could lead to surgical site infection. The surgical site infection could be caused by the spread of bacteria by immune cells [58]. Whether a therapy with probiotics can positively influence surgical site infections has not yet been conclusively clarified [59]. According to the ERAS protocol, patients who received perioperative immunonutritrion were shown to have significantly fewer infectious complications, such as surgical site infections [60]. In addition, it was shown that next-generation sequencing can help to identify culture-negative surgical side infections [61].
Firmicutes/Bacteroides ratio (F/B ratio) is also a well-studied marker for dysbiosis and is also altered during the treatment of gastrointestinal carcinoma. It has been shown that if the F/B ratio of intensive care patients is <0.1 or >10, they have a poorer outcome [62]. Thus, patients who received pelvic radiation, including colorectal carcinoma or anal carcinoma, experienced radiation-induced diarrhoea with an increased F/B ratio and dysbiosis compared with patients who did not experience diarrhoea [63]. Thus, a goal of future therapy may be that the F/B ratio improves again after treatment. For example, it has been shown that the F/B ratio is increased in obesity [64,65]. At the same time, sport leads to the reduction of the F/B ratio [66]. Whether exercise also has a positive impact on complication rates in the therapy of gastrointestinal carcinoma is unknown. However, no studies have demonstrated any effect on the F/B ratio in dysbiosis [12,67]. The extent to which the F/B ratio is also an indicator of dysbiosis in the treatment of gastrointestinal carcinoma remains to be investigated.
One way to reduce complications is to actively influence the microbiome. Probiotics are beneficial bacteria that have a positive effect on the gut microbiome. In comparison, prebiotics are dietary fibres that help beneficial bacteria to grow. Synbiotics consist of prebiotics and probiotics. It was demonstrated in a study in which colorectal carcinoma patients who underwent surgery and received probiotics had significantly fewer days to their first postoperative defecation and flatus than the placebo group. It was also shown that diarrhoea was less frequent in the probiotic treatment group [68]. In a cohort with 46 patients undergoing resection for periampullary neoplasm, a significant reduction in mortality, duration of antibiotic therapy, length of hospital stays, infections, and postoperative complications was shown with the administration of synbiotics [69]. A review investigated the effect of probiotics and synbiotics in colorectal patients undergoing surgery. It was shown that there was significantly less severe diarrhoea, postoperative infections, chemotherapy side effects, and a shorter duration of antibiotic therapy in patients treated with probiotics or synbiotics [70]. In the colorectal carcinoma mouse model, probiotics containing Pediococcus pentosaceus inhibit tumour growth and modulate gut microbiome dysbiosis towards eubiosis [71]. A Cochrane analysis could not demonstrate a relevant effect of probiotics on radiotherapy- or chemotherapy-induced diarrhoea [72].
As mentioned before, numerous factors can influence the microbiome, e.g. diet, obesity and physical activity. It has been shown that an increased BMI during laparoscopic colorectal cancer surgery [73] and preoperative obesity is associated with poorer outcomes. On the other hand, an early oral nutrition and thereby a more physiological gastrointestinal microbiome are associated with better outcomes [74]. After gastrectomy, 41.5% of patients suffer from sarcopenia, what highlights the importance of a sufficient nutritional status [75]. That diet and nutritional status affect the microbiome also shows that decreased secondary bile acids in the gut are a reasonable basis for the growth of Clostridium difficile [76]. A study with mice suffering from multiple sclerosis showed that one hour of strength exercise increased alpha diversity and abundance of the gut microbiota. Additionally, the abundance of short-chain fatty acids-producing bacteria and the F/B ratio were decreased [77]. Whether strength exercise has a similar effect on the microbiome in gastrointestinal carcinoma patients remains unclear.
Another complication of chemotherapy with irinotecan in patients with colorectal carcinoma is diarrhoea. The gut microbiome decomposes irinotecan-metabolite back to the active form, causing side effects. A new therapy, together with apple pectin, was able to inhibit metabolites responsible for diarrhoea in cell culture with human colorectal tumour cells and, at the same time, enhance the cytotoxic effect of irinotecan [78]. However, non-response to chemotherapy may also represent a complication of chemotherapy. New therapeutic approaches try to inhibit the growth of F. nucleatum by phage-guided nanotechnology and improve the first-line chemotherapy treatment [79]. Furthermore, the fact that the side effects of chemotherapy are often treated with other drugs should not be ignored. Thus, it has been shown in animal models that proton pump inhibitors, opioids and antipsychotics are associated with dysbiosis [80]. In patients, it has also been shown that opioids, and antibiotics, such as piperacillin/tazobactam, are associated with alterations of the gut microbiome and less protective bacteria such as Blautia and Lactobacillus are detectable [81]. Nevertheless, the preoperative administration of oral antibiotics could lead to significantly fewer postoperative complications in elective colorectal surgery than mechanical bowel preparation [82].
Another therapeutic option could be faecal microbiota transplant (FMT) to influence the microbiome in patients with gastrointestinal cancer under specific therapies. Currently, data on the effect of FMT in gastrointestinal carcinoma is insufficient. However, it has been shown in animal models that when the FMT donor was a patient with colorectal carcinoma and the FMT recipient mouse was germ-free, intestinal dysplasia and proliferation could be induced [83]. In another colorectal carcinoma mouse model, oral-faecal microbiota transplantation was performed during five days of FOLFOX chemotherapy and two days beyond and compared to mice that did not receive FMT. It was demonstrated, that mice treated with FMT had less severe diarrhoea and reduced mucositis, and at the same time did not induce bacteraemia. In addition, dysbiosis was improved in FMT-treated mice after FOLFOX chemotherapy [84]. From this, it can at least be concluded that the microbiome and health status of the patient can be significantly changed by FMT. Whether physiological conditions can also be achieved by FMT is unknown for gastrointestinal carcinoma. However, the fact that FMT should not be underestimated is shown by case reports in which patients have suffered severe complications or died due to FMT [85-88]. Nevertheless, FMT is a promising therapeutic option, but further studies are needed to evaluate FMT and identify the risks leading to serious side effects.
A feared complication in patients undergoing surgery on the gastrointestinal tract is sepsis, even though these are not specific complications for operations on the gastrointestinal tract. The gut microbiome also interacts with the immune system. Therefore, an altered microbiome cannot be neglected concerning the immune response. This directly influences the mucosal adaptive immunity and immune cells [89]. Meanwhile, it has been shown that the microbiome influences the course of sepsis [90]. Patients who have undergone gastrointestinal surgery and suffer from dysbiosis resulting from the procedure may have a worse outcome if they develop sepsis.
Unfortunately, there are currently insufficient data for patients with gastrointestinal carcinoma under radiation and possible complications associated with the microbiome. Therefore, additional studies with prostate and cervical cancer patients were presented here. Furthermore, it was shown that dysbiosis and associated inflammation have been found with pelvic radiation in mice, which may be an additional factor for intestinal radiation-induced damage [91].
Furthermore, it is important to remember that the microbiota does not only consist of bacteria; viruses and fungi are an additional part of the microbiota. The virome as part of the microbiota also has an impact on outcomes in patients with colorectal cancer. Thus, virome-associated risk groups could be identified [92]. The extent to which the individual viruses, fungi, and bacteria affect both the patient and each other must be investigated in more detail.
Another challenge will be that the microbiome differs depending on whether a healthy organ or a carcinoma is present, as well as the type of underlying carcinoma. For example, different microbiomes could be detected between oesophageal squamous cell carcinoma tissue and gastric cardia adenocarcinoma tissue [93]. This was confirmed in a microbiome comparison of a healthy oesophagus, Barrett's oesophagus, an oesophageal squamous cell carcinoma, and oesophageal adenocarcinoma [94].
The microbiome in colorectal carcinoma is better studied compared to oesophageal and gastric carcinoma. So far, the data for oesophageal and gastric carcinoma are limited regarding perioperative complications and complications during chemotherapy. Thus, only statements about the overall outcome or risk of recurrence with different microbiomes in oesophageal carcinoma have been made so far [31,95].
Reviewer 2 Report
The author reviews in the article "The association of gut microbiota and complications in gastrointestinal cancer therapies" the microbiota in GI cancer and their influence in therapies. The author said it is important to summarize the research interest because we need a better understanding of the microbiota and which factors trigger complications during cancer therapy and that the microbiome might be considered in the future. The review is well written and the author is summarizing the newest publications in this review for this research field. The most important points for the gut microbiota and complications in GI cancer therapies are discussed and summarized.
Author Response
Dear Reviewer,
Please find enclosed a detailed point-to-point response to the reviewers’ comments.
We hope you will find the revised version appropriate for publication in Biomedicines.
On behalf of all authors,
Yours sincerely,
Felix Schmitt, MD
Response to the Reviewers
Response to the suggestions of Reviewer 2:
Reviewer’s comment:
The author reviews in the article "The association of gut microbiota and complications in gastrointestinal cancer therapies" the microbiota in GI cancer and their influence in therapies. The author said it is important to summarize the research interest because we need a better understanding of the microbiota and which factors trigger complications during cancer therapy and that the microbiome might be considered in the future. The review is well written and the author is summarizing the newest publications in this review for this research field. The most important points for the gut microbiota and complications in GI cancer therapies are discussed and summarized.
Authors’ statement:
We appreciate the kind feedback from the Reviewer! Thank you very much!